# Upscaled production of an ultramicroporous anion-exchange membrane enables long-term operation in electrochemical energy devices

Wanjie Song[1,3], Kang Peng[1,3], Wei Xu[2], Xiang Liu[1], Huaqing Zhang[1], Xian Liang[1], Bangjiao Ye[2], Hongjun Zhang[2], Zhengjin Yang[1], Liang Wu[1] ✉, Xiaolin Ge[1] ✉ & Tongwen Xu[1] ✉

The lack of high-performance and substantial supply of anion-exchange membranes is a major obstacle to future deployment of relevant electrochemical energy devices. Here, we select two isomers (m-terphenyl and p-terphenyl) and balance their ratio to prepare anion-exchange membranes with well-connected and uniformly-distributed ultramicropores based on robust chemical structures. The anion-exchange membranes display high ion-conducting, excellent barrier properties, and stability exceeding 8000 h at 80 °C in alkali. The assembled anion-exchange membranes present a desirable combination of performance and durability in several electrochemical energy storage devices: neutral aqueous organic redox flow batteries (energy efficiency of 77.2% at 100 mA cm$^{-2}$, with negligible permeation of redox-active molecules over 1100 h), water electrolysis (current density of 5.4 A cm$^{-2}$ at 1.8 V, 90 °C, with durability over 3000 h), and fuel cells (power density of 1.61 W cm$^{-2}$ under a catalyst loading of 0.2 mg cm$^{-2}$, with open-circuit voltage durability test over 1000 h). As a demonstration of upscaled production, the anion-exchange membranes achieve roll-to-roll manufacturing with a width greater than 1000 mm.

The intrinsically intermittent nature of renewable energy (e.g., solar and wind) urgently requires electrochemical energy storage and conversion technology to improve its utilization efficiency[1–3]. Wherein, aqueous redox flow batteries, water electrolysis, and fuel cells are typically technologies for realizing electric-chemical energy conversion and storage. Among these electrochemical processes, ion conduction polyelectrolyte membranes act as charge carrier conductors and electrolyte separators, which are common and core components of these technologies[4–6]. With increasing reliance on low-cost materials, anion-exchange membranes (AEMs) possess the advantage of low manufacturing costs and allow the use of non-platinum group metal (PGM) catalysts, which are highly competitive with proton exchange membranes (PEMs)[7]. Therefore, scalable energy-effective AEMs with high ionic conductivity and stability are urgently desirable for large-scale grid energy storage and conversion applications.

Ionic conductivity plays a crucial factor for AEMs in guaranteeing the efficient operation of electrochemical devices. The inherent defect of AEMs (lower migration rate of large OH$^-$ ions) inevitably leads the

[1]CAS Key Laboratory of Soft Matter Chemistry, Collaborative Innovation Centre of Chemistry for Energy Materials, School of Chemistry and Material Science, University of Science and Technology of China, Hefei 230026, P.R. China. [2]State Key Laboratory of Particle Detectionand Electronics, University of Science and Technology of China, Hefei 230026, P.R. China. [3]These authors contributed equally: Wanjie Song, Kang Peng. ✉e-mail: liangwu8@ustc.edu.cn; gexl@ustc.edu.cn; twxu@ustc.edu.cn

conductivity below than PEMs. Increasing ion exchange capacity (IEC) is a straightforward approach for preparing high anion-conductive membranes, but this tends to reduce the mechanical strength of AEMs, which limits the applicability of AEMs in energy devices[8]. Years of research have witnessed the benefits of morphology engineering in alleviating the trade-off between ionic conductivity and mechanical robustness[4]. Nevertheless, irregular and unconnected conductive regions and extensive hydrophobic polymer matrix retard the rapid transport of ions. Encouragingly, the membrane with a sizeable inter-chain gap represented by microporous polymers is expected to alleviate the obstruction of ion movement. For instance, the first Tröger's Base polymeric AEM previously reported in our group[9] with large subnanometer free-volume voids showed a fast ion-transport rate (OH$^-$ conductivity up to 164.4 mS cm$^{-1}$ even at a low IEC of 0.82 mmol g$^{-1}$). However, unstable group and bridged bond formed during polymerization enable most current microporous polymers have the poor chemical stability, low mechanical strengthen, and poor processability. Furthermore, the good barrier property for other species besides the target ion is also challenge for microporous polymers in fitting the requirements of energy devices. Hence, the key scientific challenge for acquiring highly selective ion-transport membranes is the design of stable and robust materials while achieving precise control over pore size and distribution.

Stability is another vital index for AEMs, which determines long-term operation in electrochemical devices. Although various cationic groups[10–14] (e.g., quaternary ammonium salts, imidazole, phosphorus, guanidine, organometallic cations) explored earlier have shown substantial stability improvements, they still display inferior conductivity or long-term alkali stability. Furthermore, most previous studies are based on easily degradable AEM backbones[9,15–18] (polyphenyl ether, polyether ether ketone, polysulfone, polyolefins and Tröger's base), resulting in poor structural integrity of the AEMs. Recently, with the successful transfer of polyhydroxy alkylation and the discovery of stable piperidinium[19], highly durable AEMs can be fabricated in a simple and efficient method with the combining of ether-bond-free aryl backbone and stable cyclic quaternary ammonium group. For instance, Yan et al.[6] prepared PAP-TP-85 AEM of excellent comprehensive performance by controlling molecular weight. Lee et al.[20–22] reported a series of AEMs of high fuel cell performance by selecting monomers. Although this method can obtain AEMs with excellent

stability, the studies about the regulation of microporous morphology are relatively few. As mentioned above, constructing microporous structures within AEMs significantly facilitates ionic conduction. Therefore, the design of AEMs with microporous and stable structure is expected to achieve simultaneous improvement of the conductivity and stability of AEMs.

The m-triphenyl with "angular structure" possesses a torsion angle of 34 and 35 degrees between adjacent benzene rings, which can be employed as a twisted functional unit in the polymer chain to improve flexibility. It has been widely used for fabricating helical polymers[23] and constructing two-dimensional polymer networks[24]. As the isomer, p-triphenyl has been widely introduced into polymer chains as rigid units to enhance polymer strength[6]. A report by Mayadevi et al.[25] shows that the membrane performance is highly related to the ratio of copolymers between m- and p-triphenyl. Inspired by this, our investigation begins with combining two isomers to regulate the rigid and flexible segments by altering the ratio to control chain packing behavior and micropores constructed inside the polymer. Moreover, moderate cross-linking agents were introduced into the structure to strengthen the membrane stability. Hence, we obtained intrinsically ultramicroporous AEMs by Friedel-Crafts reaction in the absence of unstable groups and bridged bonds. The subnanometer-level intrinsic micropores can be tunable by precisely balancing the combination of m- and p-terphenyl in polymer backbones. Both experimental and molecular simulation results confirm that the AEM (MTCP-50) with the optimal ratio (1:1 of m-terphenyl to p-terphenyl) possesses narrow and well-interconnected subnanometer voids, which facilitates fast and selective anion transport within the confined channels. The resulting MTCP-50 also demonstrates robust stability owing to the chemical robust backbone and alkaline-stable piperidinium cation. These features enable the present AEM to exhibit fascinating combination of performance and durability in electrochemical devices, including neutral aqueous organic redox flow batteries, water electrolysis, and fuel cells.

## Results
### Pilot-scale manufacturing of MTCP-x polymer AEMs
The MTCP-x polymer was synthesized by two steps of superacid catalyzed polymerization and quaternization (Fig. 1). A detailed synthesis procedure and structural confirmation were provided in

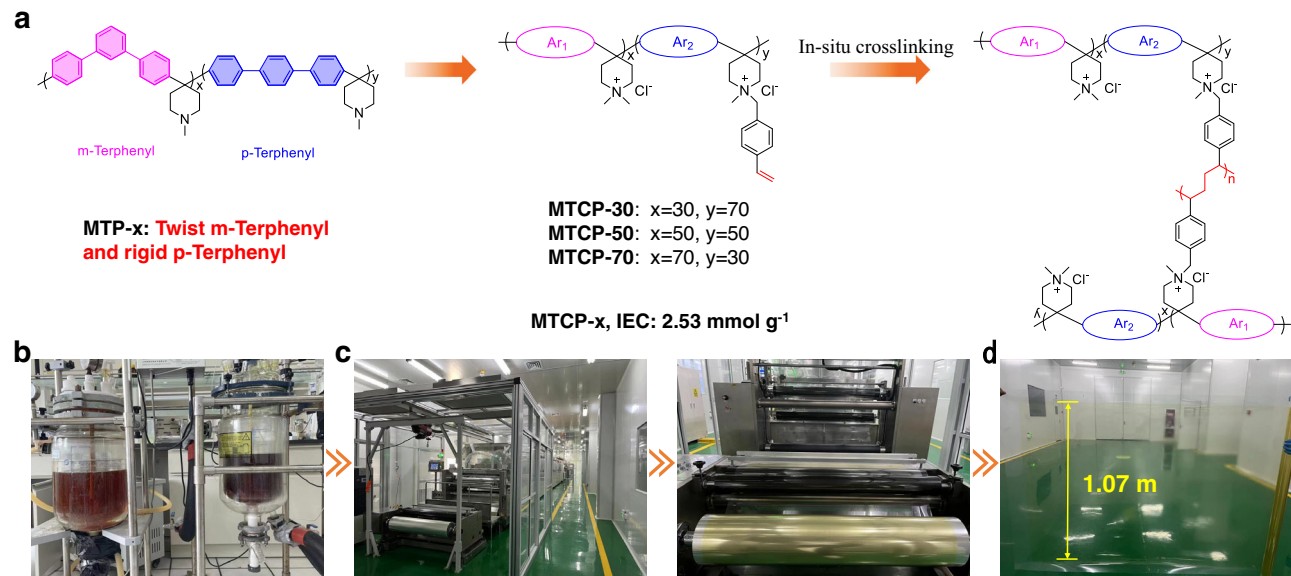

**Fig. 1 | Pilot-scale manufacturing of MTCP-x AEMs. a** The synthesis procedure of MTCP-x AEMs. **b** Photo images of MTCP-50 polymer synthesis reactor in kg-scale. **c** Photo images of the roll-to-roll membrane casting machine. **d** Photo images of transparent MTCP-50 AEM with a width >1000 mm.

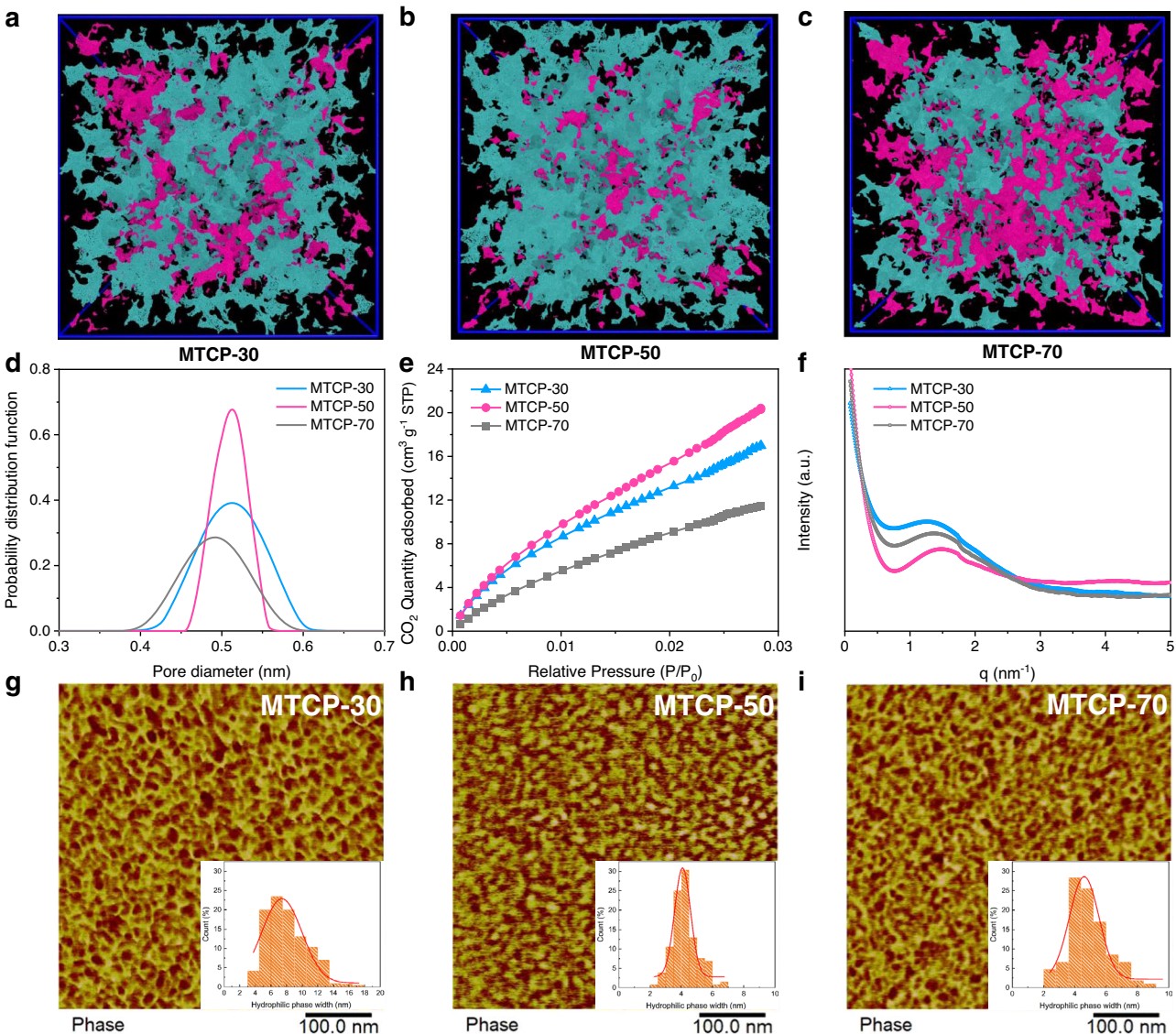

**Fig. 2 | Characterization of MTCP-x AEMs. a–c** Computational modeling results of the pore surfaces for the MTCP-x series applying a 2.0 Å probe diameter; teal and magenta indicate the accessible (interconnected) and non-accessible (disconnected) surface area, respectively. **d** Pore size distributions obtained from PALS using CONTIN analysis. **e** The $CO_2$ adsorption isotherms of MTCP-x AEM at 298.15 K. **f** SAXS patterns of MTCP-x. **g–i** AFM microphase morphology of MTCP-x (the inner images are the distribution of hydrophilic phase width of MTCP-x analyzed by the Nano Measurer).

Supplementary materials. The pilot-scale produced MTCP-x polymer (~1.5 kg) was dissolved in dimethyl sulfoxide (DMSO) to yield a 25 wt% solution. Then, as shown in the Supplementary videos, a continuous roll-to-roll membrane of 10 square meters was produced on a polyethylene terephthalate (PET) substrate. The photograph in Fig. 1d shows that the fabricated AEM has excellent film-forming properties and can easily be made into thin and transparent membranes over large areas (width >1000 mm).

MTCP-x AEMs with the same IEC (2.53 mmol g⁻¹) were prepared via a precisely controlled ratio of p-terphenyl and m-terphenyl (where x stands for the molar ratio of m-terphenyl) to elucidate the structure-performance relationship. Firstly, molecular simulation was adopted to construct the polymer structure model. The generated model structures suggest that the inefficient packing of molecular chains caused by rigid and twisted units produces ultra-microcavity[26]. We further conducted positron annihilation lifetime spectroscopy (PALS)[27] to characterize the microporosity of materials, and elucidate the size, content and distribution of microporosity through free volume voids capturing and annihilation process of positronium in the material. Fig. 2a–c display

that MTCP-70 with more twist monomer exhibits more disconnected microporous cavities (magenta) and less micropore content (Supplementary Fig. 6i). As the content of rigid monomer increases to 50% (MTCP-50), the proportion of interconnected microporous cavities (teal) and micropore content increased significantly. However, further increasing the ratio of rigid monomers (MTCP-30), the micropore content only increased slightly. But some larger pores were generated due to the more inhibitory ability of chain packing efficiency, which may be detrimental to the barrier property. Moreover, the pore size distribution simulations show that the MTCP-50 has the narrowest pore size distribution (Supplementary Fig. 7a). This coincides well with the microporosity characterization obtained from PALS (Fig. 2d). Accurately speaking, we can't obtain thoroughly evidence of micropore connectivity from the PALS experiment. Therefore, we performed $CO_2$ adsorption, which defines the interconnected microporosity accessible by $CO_2$ at 298.15 K[28,29]. As shown in Fig. 2e, MTCP-50 has the largest adsorbing capacity, suggesting that there are more interconnected micropores within MTCP-50, which is consistent with the simulation results. These results confirmed that adjusting the ratio of p-terphenyl

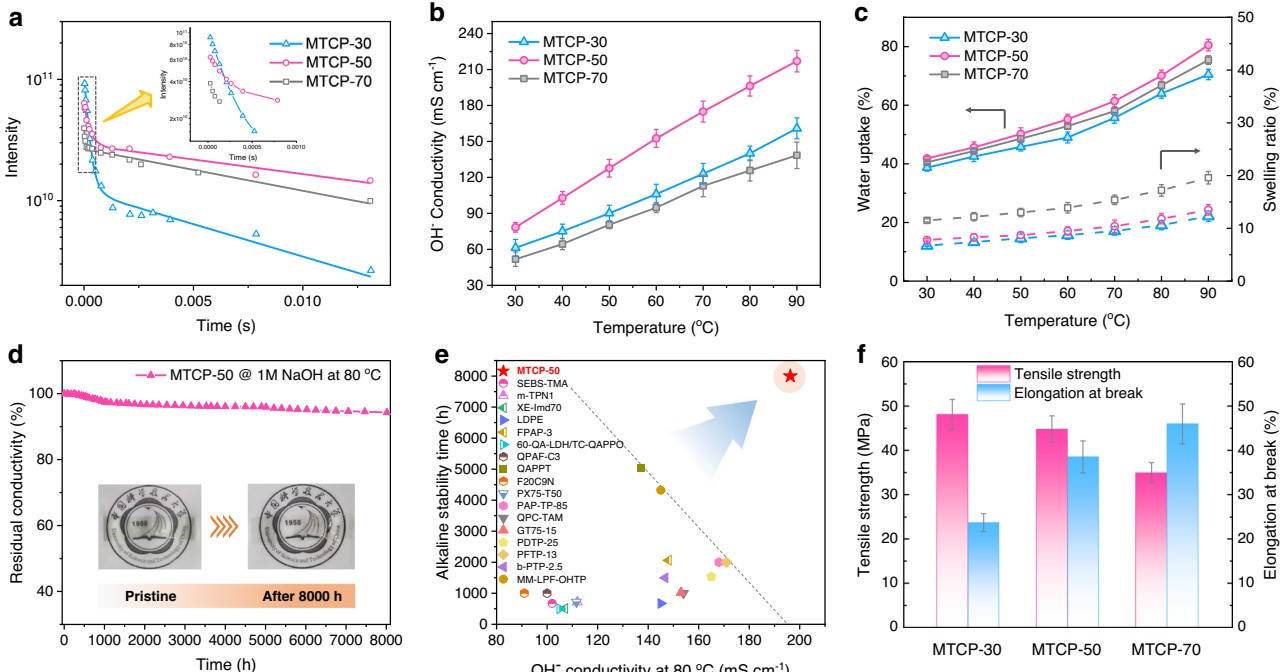

**Fig. 3 | Membrane relevant properties. a** The analysis of $T_2$ from the $^{35}$Cl ssNMR of MTCP-x. **b** Temperature-dependent OH$^-$ conductivity of MTCP-x. The error bars represent the standard deviation of OH$^-$ conductivity. **c** Water uptake and dimensional swelling versus temperature. The error bars represent the standard deviation of water uptake and swelling ratio. **d** Alkali stability with respect to OH$^-$ conductivity of MTCP-50 in 1 M NaOH at 80 °C (Note that stability is not the end-of-life data but of test data). **e** Summary of the relationship between OH$^-$ conductivity and alkaline stability (recent research progress for AEMs of OH$^-$ conductivity over 90 mS cm$^{-1}$ at 80 °C and ex-situ alkaline stability exceeding 500 h. Refer to Supplementary Table 4). **f** TS and EB of MTCP-x in OH$^-$ form at room temperature (dry state). The error bars represent the standard deviation of mechanical properties.

and m-terphenyl can lead to different pore sizes and distributions. Among the three MTCP-x-based membranes, both simulation and experimental results reveal that MTCP-50 has the most extensive and interconnected voids. More importantly, the MTCP-50 displays a narrow density function, demonstrating the uniform distributions of voids. This anomalous phenomenon illustrated the regular and periodic molecular chain under a unique balance. The narrow distribution and well-connected ultramicropores in MTCP-50 are expected to enable fast selective ion transport.

Morphology control is another vital factor for ion conduction. Small-angle X-ray spectroscopy (SAXS) and atomic force microscopic (AFM) are powerful evidence for investigating phase-separated morphology. As shown in Fig. 2f, the detected obvious scattering peaks in SAXS profiles suggest the phase separation of MTCP-x AEMs and determine the length scale of the ion channel. The enhanced peak intensity in MTCP-50 indicates a more well-developed interconnected network between the hydrophilic channels. Additionally, the smaller *d*-spacing of MTCP-50 between MTCP-x enables enhanced selectivity for redox-active molecules. Further, the alternating arrangement of bright (hydrophobic polymer backbone domains) and darker (hydrophilic ionic domains) in AFM images (Fig. 2g–i) support the phase separation morphology. Compared with MTCP-30 and MTCP-70 (more dead channels and poor channel size uniformity), MTCP-50 exhibits a more-ordered morphology. Importantly, the analysis of hydrophilic ionic domains measured by Nano Measurer demonstrates more uniform distribution of ion channels within MTCP-50. Both the microporosity and morphology characterization indicate that the MTCP-50 with 50% molar ratio of m-terphenyl is the optimal value, which is consistent with the research result of Mayadevi et al. The evenly distributed and interconnected ion channels within MTCP-50 arose from the well-balanced spatial arrangement of the polymer backbone between m- and p-terphenyl, which will facilitate efficient ion transport. Moreover, the cross-linking does not shift the optimization value of 50% and is

expected to advance membrane performance in dimensional and chemical stability.

## Membrane relevant properties

The structural features demonstrated for MTCP-x reveal corresponding membrane properties. The solid-state nuclear magnetic resonance (ssNMR) experiments reveal the motion strength of ions by the analysis of spin-spin relaxation time ($T_2$)[30]. The longer $T_2$ suggests the less confined motion of ions in membrane. Among MTCP-x, MTCP-50 exhibits a longer $T_2$ (Fig. 3a, 0.20 ms of MTCP-50 *vs.* 0.18 ms of MTCP-30 and 0.05 ms of MTCP-70) and narrower signal (Supplementary Fig. 9), demonstrating the more interconnected voids and well-order ion channels within MTCP-50 enable fast anion transport in membrane. As expected, the MTCP-50 exhibits the highest OH$^-$ conductivity of 78.4 mS cm$^{-1}$ at 30 °C, drastically enhancing to 217.0 mS cm$^{-1}$ with temperature elevated to 90 °C (Fig. 3b). For comparison, the conductivity is not ideal for either MTCP-70 (small pore size, more disconnected pore distribution, and less-ordered ion domains) or MTCP-30 (uniformly distributed pore and ion domains). Additionally, the MTCP-50 shows advantageous in water absorption behavior (~41.9%, Fig. 3c) and anti-swelling (<8%) at 30 °C because large and highly uniform voids, as well as well-ordered ion domains, can evenly disperse the water molecules. The MTCP-30 with more rigid elements exhibits hydrophobic characteristics, coupled with more voids resulting in a lower water uptake (~38.7%) and swelling degree (6.7%). The MTCP-70 with more twist elements increased membrane flexibility and further caused the membrane tends to be tightly stacked. Thus, the resulting high chain mobility and fewer voids make the membrane exhibit poor anti-swelling behavior (11.5%).

Membrane stability in harsh conditions is crucial for the practical operation of electrochemical devices. Figure 3d plots the OH$^-$ conductivities loss rate of MTCP-50 by alkali aging in 1 M NaOH solution at 80 °C. After systematically testing longer than

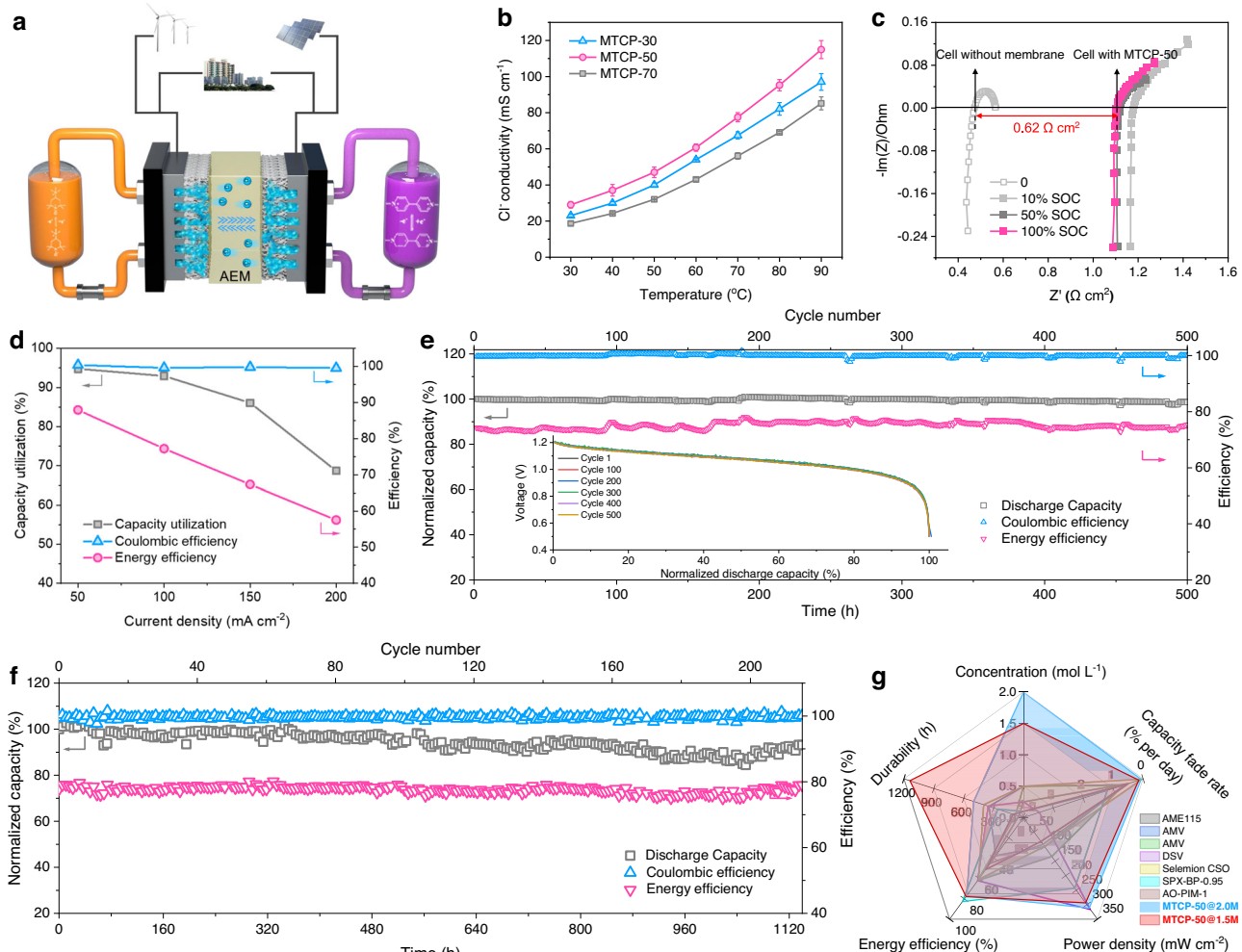

**Fig. 4 | MTCP-50 based AORFB performance. a** Schematic diagram of the TEMPTMA/MV cell assembled with the MTCP-50 AEM ($54 \pm 2\,\mu m$) and the conduction of $Cl^-$ ions across the membrane matrix. **b** $Cl^-$ conductivity versus temperature. The error bars represent the standard deviation of $Cl^-$ conductivity. **c** Electrochemical impedance spectroscopy (EIS) spectra measured in cells assembled with MTCP-50 at varied SOC. **d** Capacity utilization, CE and EE of MTCP-50-based cells at varied current densities. **e** Long-term galvanostatic cycling of a MV/TEMPTMA cell assembled with MTCP-50 at $100\,mA\,cm^{-2}$. (Pumped $5\,cm^2$ test cell. The posolyte comprises 5 mL of 2.0 M TEMPTMA while the negolyte comprises

7.5 mL of 2.0 M MV. The cutoff voltages are 1.6 V and 0.5 V, and a potential hold is applied until the current density falls below $4\,mA\,cm^{-2}$). **f** Long-term galvanostatic cycling for 1.5 M MV/TEMPTMA cell assembled with MTCP-50 (Experiments: pumped $50\,cm^2$ test cell; 1.5 M solutions of TEMPTMA and MV, $100\,mA\,cm^{-2}$). **g** A radar plot comparing the electrolyte concentration, long-term durability, capacity fade rate, EE (at a current density of $100\,mA\,cm^{-2}$) and PPD of MTCP-50 and representative AEMs and PEMs-based AORFBs. A more comprehensive and detailed AORFB performance comparison is provided in Supplementary Table 5.

8000 h, the MTCP-50 displays ~94.3% ionic conductivity retention. Additionally, the membrane keeps transparent and maintains its mechanical toughness (Supplementary Fig. 18). The Raman spectral shows no observable group degradation (Supplementary Fig. 15). As confirmed by the $^1H$ NMR spectrum (Supplementary Fig. 16), the ~5.7% conductivity loss probably results from minor degradation of ring opening and nucleophilic substitution of piperidinium cation. To the best of our knowledge, it is the longest record of alkaline stability and is superior to other AEMs in terms of $OH^-$ conductivity and alkali stability[6,20,21,31-41] (Fig. 3e). Furthermore, the MTCP-50 displays remarkable oxidative stability (Supplementary Fig. 19, ~97.74% weight retention after oxidative stability test at 80 °C for 24 h). This impressive membrane stability is attributed to the coordination action of the flexible backbone (partially flexible m-triphenyl reduces the restriction on ring strain relaxation) and cross-linked structure. Additionally, the mechanical properties in Fig. 3f shows that the MTCP-50 has an excellent tensile strength (TS, 44.8 MPa) and elongation at break (EB, 38.5%). All these

provide an essential guarantee for long-term operation in various electrochemical devices.

## Neutral aqueous organic redox flow batteries (AORFBs) performance

AORFBs, as a new energy storage technology, is a powerful tool to integrate renewable energy into large-scale electricity storage. This technique requires a membrane with fast ion conduction and excellent redox-active molecules impermeability. The MTCP-50 with microporous structure constructed via precisely regulated monomers delivers a $Cl^-$ conductivity approaching $29\,mS\,cm^{-1}$ at 30 °C (Fig. 4b). Most importantly, the narrow distribution of ultramicropores, with an average diameter of 5.2 Å (much smaller than the size of redox-active molecules, Supplementary Fig. 22) is an excellent barrier to redox-active molecules, which was confirmed by the concentration-driven diffusion experiments of N,N,N-2,2,6,6-heptamethylpiperidinyl oxy-4-ammonium chloride, TEMPTMA and Methyl viologen dichloride, MV (Supplementary Fig. 25). Hence, the TEMPTMA/MV cells using MTCP-50 as electrolyte separator and $Cl^-$ transport carrier were configured to assess the performance. As

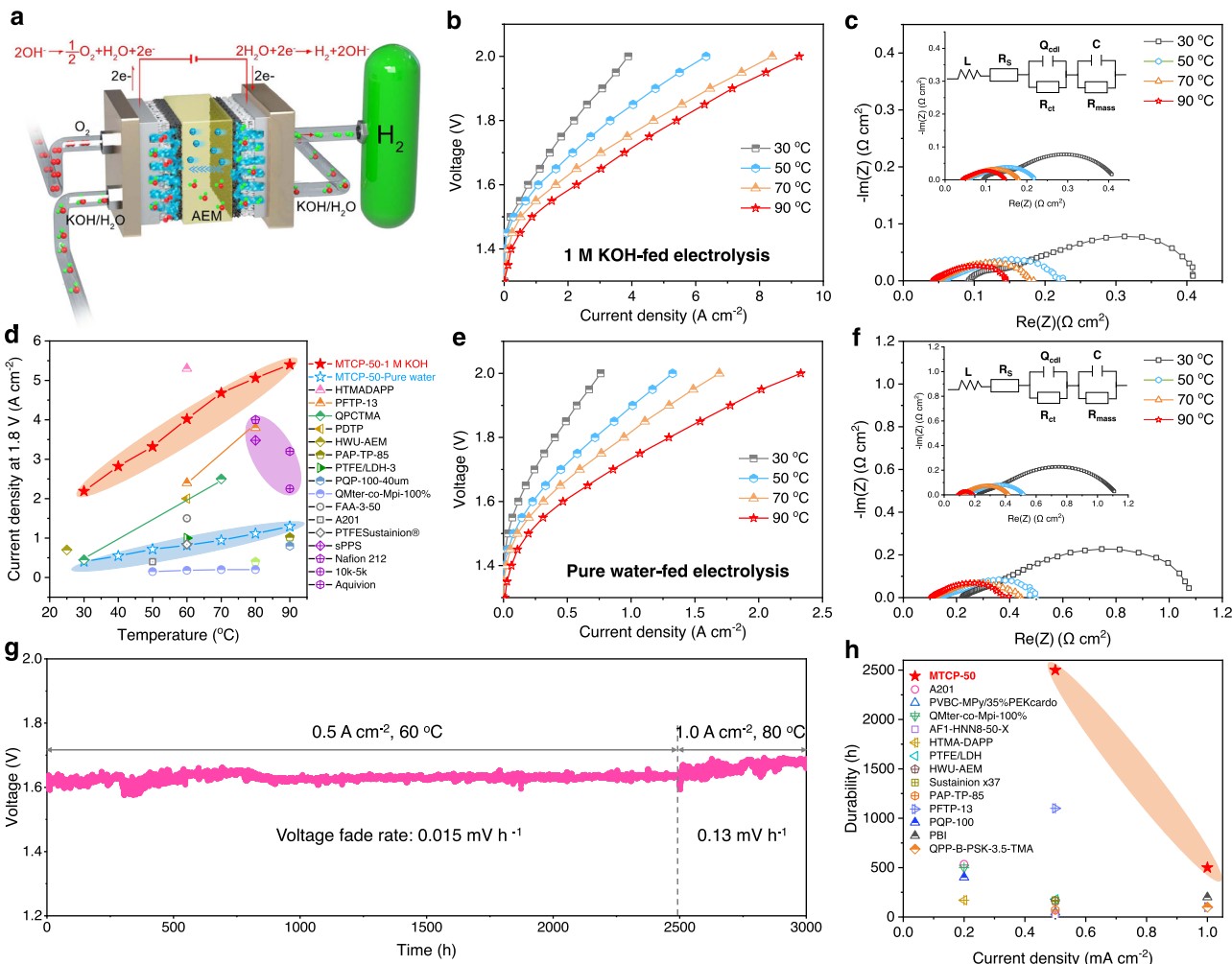

**Fig. 5 | MTCP-50 based AEMWEs performance. a** Schematic diagram of AEMWEs assembled with the MTCP-50 AEM ($25 \pm 2\,\mu m$) and the conduction of $OH^-$ ions across the membrane matrix. **b** $I-V$ curves of versus temperature with 1 M KOH feed. **c** EIS spectra of AEMWEs versus temperature under 1 M KOH feed. **d** Summary of current densities at 1.8 V of current AEMWEs (MTCP-50 fed with 1 M KOH, pink; MTCP-50 fed with pure water, blue; proton exchange membrane, purple). **e** $I-V$ curves of versus temperature with pure water feed. **f** EIS spectra of AEMWEs versus temperature under pure water feed. **g** Long-term durability of AEMWEs under $0.5\,A\,cm^{-2}$ at 60 °C for 2500 h and $1.0\,A\,cm^{-2}$ at 80 °C for 500 h. **h** Comparison the durability of 1 M KOH-fed alkaline electrolyzers with current state-of-the-art AEMWEs (Refer to Supplementary Table 6).

depicted in Fig. 4c, the cell running at 2.0 M electrolyte concentration reveals a lower area-specific resistance of $0.62\,\Omega\,cm^2$ by subtracting the contributions from other components. And it also presents an exceptional peak power density (PPD) of $313.7\,mW\,cm^{-2}$ (~100% SOC, Supplementary Fig. 27b), which is ascribed to the low area-specific resistance contributing from the highly anion-conductive MTCP-50. Such high anion transport of MTCP-50 also enables the configured cell to display an current rate performance over the range of $50-200\,mA\,cm^{-2}$. Figure 4d displays that it can operate at high current density, the capacity utilization, coulombic efficiency (CE), and energy efficiency (EE) of the cell are 92.9%, 99.6%, and 77.2% at $100\,mA\,cm^{-2}$, respectively. Even the current density up to $200\,mA\,cm^{-2}$, it achieves a high capacity utilization (68.7%), CE (99.6%), and EE (57.5%), making fast charging possible. These results indicate the superior performance of MTCP-50-based cells among reported pH-neutral AORFBs (Supplementary Fig. 32) and suggest that high anion-conductive AEM plays an essential part in the efficient operation of AORFBs.

Galvanostatic cycling performance based on MTCP-50 under high electrolyte concentration of 2.0 M was further performed. Figure 4e plots the representative discharging profiles and cycling data. Even after continuous running for 500 h, the cell delivers a total capacity retention of 99.9974% per hour. The average CE and EE remains at

~99.9% and ~76.4%, respectively. Both the post-cycling cyclic voltammogram (Supplementary Fig. 29) and $^1H$ NMR spectra (Supplementary Figs. 30 and 31) measurements show no crossover °Ccurred for redox-active molecules, further confirming the high selectivity of MTCP-50 and extraordinary stability of AORFBs. To further approach the actual application scenario, we scaled up the area of MTCP-50 ($50\,cm^2$) assembled in AORFBs. The capacity was also scaled up with the increase in electrolyte volume. In the cycling test of 1140 h using 1.5 M electrolyte concentration, a high CE (~100%) and EE (~77.8%) can be kept at an unattenuated level. The cell retains its original discharge capacity, equivalent to a capacity retention of 99.9939% per hour (Fig. 4f). By comparing the comprehensive performance[28,29,42–46] (Fig. 4g), it is found that the MTCP-50-based AORFB outperforms current advanced AORFBs assembled with other AEMs and PEMs in terms of cycle time, cycling stability, EE and PPD even at high concentration. This remarkable performance enables the MTCP-50 to be reliable material for the AORFBs application.

## Anion-exchange membrane water electrolyzers (AEMWEs) and fuel cells (AEMFCs) performance

We have demonstrated the advantages of the prepared membrane used for energy storage in AORFBs under room temperature. The

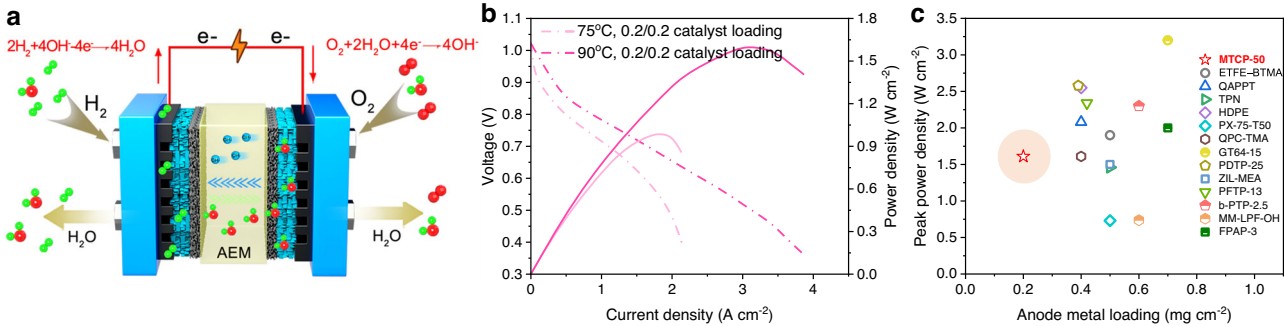

**Fig. 6 | MTCP-50 based AEMFCs performance. a** Schematic diagram of AEMFCs assembled with the MTCP-50 AEM ($25 \pm 2\,\mu m$) and the conduction of OH⁻ ions across the membrane matrix. **b** The performance of AEMFCs at different temperatures. **c** Summary of the PPD versus the noble-metal loading (Refer to Supplementary Table 7).

application potential in hydrogen energy community, e.g., water electrolysis and fuel cell, which operate at medium-high temperatures, were further evaluated. Undoubtedly, increasing operating temperature is conducive to improved ionic conductivity and electrochemical kinetics. However, the insufficient properties, including inferior alkaline stability and anti-swelling properties at high temperatures of most existing AEMs, limit the working temperature of current AEMWEs & AEMFCs below 80 °C, which hinders efficient hydrogen production or electricity. The conductivity tests in Fig. 3b suggest the more enhanced ionic conduction capacity of MTCP-50 when increasing temperature. Additionally, over 8000 h of alkali resistance stability testing under 80 °C has proven the excellent stability of the MTCP-50 AEM. More importantly, the still low swelling ratio at high temperatures avoids the problem of ion conductivity and mechanical properties degradation caused by excessive water absorption. These features enable MTCP-50 to have excellent application prospects in AEMWEs & AEMFCs.

We first investigated the electrolysis performance of AEMWE based on MTCP-50. As shown in Fig. 5b, when electrolyzers fed 1 M KOH solution, we achieved impressive current density, ~2.2 A cm⁻² at 1.8 V and 30 °C. The high performance is mainly attributed to two aspects. One part, the fast transfer rate of OH⁻ reduced the ohmic impedance (Fig. 5c, $0.044\,\Omega\,cm^2$). Another part, the high water diffusivity (Supplementary Fig. 14), from anode to cathode, promoted the mass transfer efficiency[21]. The enhanced ionic conductivity and high-temperature tolerance properties of MTCP-50 ensure the present MTCP-50-based AEMWE can still yield a stable high energy output (5.4 A cm⁻² at 1.8 V) even running at 90 °C. From the current densities summarized in Fig. 5d for the state-of-the-art AEMWE at 1.8 V[2,39,47–60], we can find that most of the AEMWEs display performance below 2 A cm⁻², while commercially available PEMWEs have performance above 3 A cm⁻². Compared with recently reported AEMWEs and PEMWEs, our present AEMWEs display an advanced level. Considering pure water-feeding AEMWEs have attracted more attention for their ability to avoid lye corrosion and promising commercialization. The MTCP-50-based AEMWE is then conducted on a pure feedwater system and also delivers an excellent performance in Fig. 5e, e.g., 1.3 A cm⁻² at 1.8 V and 90 °C. It is attributed to the fast ion-transport rate of the membrane itself, which can be reflected in the ohmic impedance data (Fig. 5f, $0.107\,\Omega\,cm^2$).

The long-term durability is essential for AEMWEs. Figure 5g presents that the MTCP-50-based AEMWE can operate under 0.5 A cm⁻² for 2500 h at 60 °C with only a 0.015 mV h⁻¹ voltage decay rate. Then we increased the current density and continued to test. Notable, even at a rigorous condition of 1.0 A cm⁻² and 80 °C, the MTCP-50-based AEMWE can still operate for 500 h and maintains a minor fading rate of 0.13 mV h⁻¹. Such more extended durability profits from the highly alkali-resistant MTCP-50 in harsh environments. Our current AEMWE shows the highest current density and long-term durability in 1 M KOH

(Fig. 5h)[2,47,49–52,55,61–65], which indicates that the MTCP-50 has broad application prospects in AEMWEs.

For AEMFCs, future development will be towards low-loading platinum or non-platinum catalysts. Increasing temperature has the additional benefit of better water management and reducing carbonation, etc., making it possible for AEMFCs to achieve performance improvement independent of catalyst load[6,66]. The MTCP-50 assembled AEMFC shows a high PPD of 1.61 W cm⁻² at 90 °C (Fig. 6b). This performance is obtained at a low PGM loading of 0.2 mg cm⁻² for both cathode and anode, which is superior to the performance under 75 °C (1.0 W cm⁻²). This low PGM catalyst load dependence performance may be attributed to accelerated ion transport and increased catalyst activity at high temperatures. Figure 6c summarizes the recent reports of the PPD as a function of the anode metal loading. It indicates that MTCP-50-based AEMFC shows prevail over the state-of-the-art AEMFCs[14,21,22,39,41,66–72]. Further, a 0.077 mV h⁻¹ voltage fading rate in the open-circuit voltage hold test over 1000 h indicates the excellent chemical stability of MTCP-50. Despite the superior conduction and stability of MTCP-50 AEM, multiple factors, including membrane electrode assembly, fuel cell testing parameters, et al., affected fuel cell performance. Optimizing fuel cell testing to achieve better performance of the MTCP-50-based AEMFC will be carried out for future research.

## Discussion

In summary, we selected two isomers of terphenyl with different conformations, m-terphenyl and p-terphenyl, to prepare intrinsically ultramicroporous AEMs by Friedel-Crafts reaction. The resulting membrane shows high ion-selectivity transport rates and robust chemical and dimensional stability, making it suitable for a wide range of applications (e.g., AORFBs, AEMWEs, AEMFCs) and demonstrating excellent performance. We envision that the concept established here will help advance the development of ionic membrane design.

## Methods
### Materials
P-Terphenyl, 1-methyl-4-piperidone, Trifluoroacetic acid (TFA), 4-Vinylbenzyl chloride (VBC), iodomethane (CH₃I), trifluoromethanesulfonic acid (TFSA) were obtained from Energy Chemical. M-Terphenyl was purchased from Aladdin. Dichloromethane (CH₂Cl₂), potassium hydroxide (KOH), potassium carbonate (K₂CO₃), DMSO, sodium hydroxide (NaOH), sodium chloride (NaCl), and acetone were acquired from Sinopharm. The polymerization inhibitor needs to be removed before VBC application, and no further purification is required for other chemicals.

### Synthesis of MTCP-x polymers
MTCP-50 was synthesized according to a typical procedure: p-Terphenyl (4.6 g, 20 mmol) and m-Terphenyl (4.6 g, 20 mmol) were

first added into flask surrounded with ice bath. Subsequently, 1-methyl-4-piperidone (4.97 g, 48 mmol) and CH$_2$Cl$_2$ (4 mL) were then quickly infunde the cooling flask. TFA (1.6 mL) and TFSA (20 mL) were then dropwise. The mixture was continued to react at 0 °C for 9 h. Then, the very sticky solution was poured into NaOH aqueous solution (aq., 2 mol L$^{-1}$, 1000 mL). Stirred vigorously for 36 h and washed with water for 5 times. Followed by filtrating to obtain the product. Finally, the fibrous copolymer was dried at 60 °C to obtain MTP-50. Then, a one-neck reactor was charged with MTP-50 (2.6 g, 4 mmol), K$_2$CO$_3$ (0.33 g, 2.4 mmol), VBC (0.122 g, 0.8 mmol), and 18 mL DMSO. After 24 h stirring under room temperature, excessive CH$_3$I was injected into the mixed solution for another 12 h stirring with tinfoil parceling to avert light. The resulting homogeneous solution was precipitated in acetone followed by washing with water and air-drying. The obtained product was abbreviated as MTCP-x (x = 70, 50, 30), where x is the molar ratio of m-Terphenyl.

## Membrane preparation

The membrane was prepared by casting a 25 wt% solution (~1.5 kg MTCP-x dissolved in DMSO) on a PET substrate via roll-to-roll membrane machine (speed of 0.3 m min$^{-1}$ and dried at 80 °C). MTCP-x samples in OH$^-$ or Cl$^-$ form were acquired by soaking in 1 M NaOH or NaCl aq. for 24 h. The residual solution was rinsed with DI water to remove it.

## Characterizations

$^1$H nuclear magnetic resonance ($^1$H NMR, Supplementary Figs. 2, 16, 30, 31) spectra were recorded on Bruker 510 instrument in standard solvent of d6-DMSO to confirm the chemical structure. The ssNMR (Fig. 3a, Supplementary Fig. 9) was conducted on a Bruker AVANCE NEO 600 WB spectrometer. The carbon dioxide (CO$_2$) sorption (Fig. 2e) was performed on ASAP2020M + C (Micromeritics) at 273.15 K. In situ Fourier transform infrared spectroscopy (in situ FTIR, Supplementary Fig. 5) was conducted on Thermo Nicolet iS10. Scanning electron microscope (SEM, Supplementary Fig. 4f) was conducted on GeminiSEM 500 instrument (Carl Zeiss AG) to observe the surface morphology of MTCP-50. Dynamic mechanical analysis (DMA, Supplementary Figs. 10, 11, 18) was employed on TA Instrument Q800 to obtain the stress-strain curves. Additionally, the rheological properties of MTCP-x were recorded on DMA. Specifically, all membrane samples with 1 × 4 cm were fixed in the tension clamps. Thermogravimetric analysis (TGA, Supplementary Fig. 12) was tested on TA Instruments Q5000IR with a ramp rate of 10 °C min$^{-1}$. Atomic force microscopy (AFM, Fig. 2g–i) detected the surface morphology on MultiMode V (Veeco) with tapping mode. Transmission electron microscopy (TEM, Supplementary Fig. 20g–i, 36) was used to observe the phase separation morphology and catalyst morphology on JEM 2100 F (JEOL Ltd). Dynamic vapor sorption (DVS, Supplementary Fig. 14) measured the water sorption behavior at different RHs on Aquadyne DVS (Quantachrome Ins). Raman spectra (Supplementary Fig. 15) of dry MTCP-50 AEM after alkali aging at different times was recorded on LabRamHR Evolution with the excitation wavelength λ = 785 nm. Small-angle X-ray scattering (SAXS, Fig. 2f) data were recorded on Anton Paar Saxesess mc2 instrument that employs a copper Kα. Positron annihilation lifetime spectroscopy (PALS, Fig. 2d, Supplementary Fig. 8) was employed to observe the free volume of bulk polymer MTCP-x samples. The PAL spectra was collected by a fast-fast coincidence spectrometer with a time resolution of ~200 ps. The positron source ($^{22}$NaCl, 30 μCi) was sandwiched between two identical membrane samples (1.5 cm × 1.5 cm) with a thickness exceeding 1 mm (this thickness ensures that almost all positrons are sufficient to annihilate in the sample) by stacking multiple pieces of the same membranes. The sample-source-sample was placed in a vacuum chamber. The signals of start and stop were recorded by two perpendicularly positioned scintillation detectors. The distance between the sample-source-sample set and each

lifetime detector is ~20 mm. Each spectrum was collected for 4096 channels with a channel width of 12.66 ps/channel. A total of $4 \times 10^6$ counts were collected for each PAL spectrum, with a count rate of about 600 CPS. The obtained lifetime spectra were analyzed using Lifetime 9 and CONTIN program. The o-Ps lifetime ($\tau_3$) corresponding to average radius (R), o-Ps intensities (I) and relative fractional free volume (FFV) were analyzed by below equations:

$$\tau_3 = \frac{1}{2} \times \left[ 1 - \frac{R}{R + \Delta R} + \frac{1}{2\Pi} \sin\left(\frac{2\Pi R}{R + \Delta R}\right) \right]^{-1} \quad (1)$$

$$FFV = CV_f I_3 = C\left(\frac{4}{3}\Pi R^3\right) I_3 \quad (2)$$

Where ΔR is 0.1656 nm, representing the electron layer thickness, which is obtained by empirical calibration.

## Computational details

Atomistic molecular dynamics simulations have been performed in the GROMACS[73] (version 2020.6) simulation package using the General Amber force field (GAFF2)[74]. All models were constructed by randomly placing 108 chains of polymers consist 10 monomer units into a simulation box of ~100 Å. The cross-linker molecules were also randomly inserted and the systems were equilibrated under the NPT ensemble for 20 ns before the cross-linking process. The cross-linking bonds were generated between reactive atoms on different monomer units within a cutoff of 6 under the same NPT ensemble at 600 K for 10 ns using a timestep of 0.5 fs. After the cross-linking reactions, the final structures were annealed through a multistep tempering method to the target temperature of 298 K. The accessible volume and surfaces were analyzed using the Zeo++ program with a probe diameter of 0.2 nm[75]. The Nose-hoover and Parrinello-Rahman method was used for both temperature and pressure coupling. For the nonbonded interactions, a cutoff length of 1.2 nm was implemented. And for the long-range electrostatic interactions, the Particle Mesh Ewald method[76] with a Fourier spacing of 0.1 nm was applied. All covalent bonds with hydrogen atoms were constrained using the LINCS algorithm[77].

## IEC

The substantial IEC value was identified by Mohr titration. Typically, thoroughly dry MTCP-x samples of about 100 mg in Cl$^-$ form were weighted ($W_{dry}$) and ion-exchanged in NaCl (1 mol L$^{-1}$, 60 °C, 24 h). Afterward, it was rinsed with DI water repeatedly. Subsequently, the samples were immersed in Na$_2$SO$_4$ (0.5 mol L$^{-1}$, 60 °C, 24 h) for another ion exchange. Lastly, collected the Na$_2$SO$_4$ aq. and titrated with standard AgNO$_3$ aq. (0.1 mol L$^{-1}$) K$_2$CrO$_4$ is the indicator. The IEC value in OH$^-$ form was obtained by correcting the Cl$^-$ and OH$^-$ mass differences between Cl$^-$ and OH$^-$. The consumed volume of AgNO$_3$ aq. ($V_{AgNO_3}$) was monitored and calculated the IEC as follows:

$$IEC(mmol/g) = (V_{AgNO_3} \times C_{AgNO_3})/W_{dry} \quad (3)$$

## WU and SR

The WU of MTCP-x membranes was measured in Cl$^-$ and OH$^-$ forms. Weight, length and thickness of dry MTCP-x samples with Cl$^-$ form were recorded ($W_{dry,Cl}$, $L_{dry,Cl}$, $T_{dry,Cl}$). The weight in OH$^-$ form ($W_{dry,OH}$) was obtained using IEC correction method for mass differences between Cl$^-$ and OH$^-$. Then, the MTCP-x samples were immersed in NaOH aq. (1 mol L$^{-1}$) for 12 h at different temperatures. Residual NaOH aq. was washed by DI water for 12 h. The wet weight, length and thickness ($W_{wet,OH}$, $L_{wet,OH}$, $T_{wet,OH}$) of resulting solvated MTCP-x samples were measured after wiping the excess surface water. Each

measurement was performed three times using different MTCP-x samples for precise results. The WU and SR were finally calculated from below equation:

$$Water\ Uptake(\%) = \frac{W_{wet} - W_{dry}}{W_{dry}} \times 100\% \qquad (4)$$

$$Swelling\ Ratio(\%) = \frac{L_{wet}(or\ T_{wet}) - L_{dry}(or\ T_{dry})}{L_{dry}(or\ T_{dry})} \times 100\% \qquad (5)$$

### Ionic conductivity

The in-plane anion conductivity of MTCP-x samples on a four-electrode platinum electrode was measured by AC impedance analyzer (Zahner Zennium E). The width (w) and thickness (d) of the AEM sample in $Cl^-$ or $OH^-$ form were quickly measured by graduated scale and micrometer caliper, respectively. Then fixed it in a measurement cell (the distance (L) between the two potential sensing electrolytes is 1 cm) and thoroughly immersed in DI water. The resistance (R) at a set temperature was collected over the frequency range from 1 M Hz to 100 Hz. The conductivity ($\sigma$) was calculated from:

$$\sigma = \frac{L}{Rwd} \qquad (6)$$

### Ex-situ durability

The MTCP-50 samples were soaked in NaOH aq. (1 mol L$^{-1}$) and sealed in a glass bottle at 80 °C for controlled periods (Refreshed the alkaline solution and glass bottle once a month). The changes in conductivity were monitored.

### Cyclic voltammetry (CV)

A three-electrode system consisting of a glassy carbon working electrode, standard Ag/AgCl reference electrode, and platinum coil counter electrode was employed to record CV in 1.0 M NaCl aq. using ZAHNER ZENNIUM E electrochemical workstation.

### AORFB testing

The MTCP-50 was sandwiched by three stacked sheets of carbon paper (backed SGL 39AA) to fabricate the MEA. For normal tests, TEMPTMA (2 mol L$^{-1}$, 5 mL) for catholyte and MV (2 mol L$^{-1}$, 7 mL) for anolyte were cycled by a peristaltic pump. The TEMPTMA/ MV AORFB was also tested at lower concentration with TEMPTMA (0.5 mol L$^1$, 5 mL) in NaCl aq. (1.5 mol L$^{-1}$) and MV (0.5 mol L$^{-1}$, 7 mL) in NaCl aq. (1.5 mol L$^{-1}$). Galvanostatic cycling was performed at varied current densities of 50, 100, 150 and 200 mA cm$^{-2}$ in the voltage range of 1.6–0.6 V. Polarizations were recorded on a Bio-Logic BCS-815 at certain charge (SOC). The long-term cycling test under electrolyte concentration of 2.0 or 1.5 mol L$^{-1}$ was performed at 100 mA cm$^2$. All operations of NAORFB tests were performed at room temperature under Ar atmosphere.

### AEMWE testing

The MEA was assembled with NiFe catalyst, MTCP-50, and Pt/Ru/C catalyst. Typically, the NiFe catalyst for OER electrode was prepared according to the previous reports[2]. The NiFe or Pt/Ru/C catalyst inks were evenly mixed with 20 wt% MTCP-50-0% (without grafted VBC) or 20 wt% MTCP-50 ionomer in solution of isopropanol/DI water (4 to 1). After thoroughly dispersed, the inks were sprayed on fluid collectors (5 cm$^2$; anode, Ni foams; cathode, Toray 060). The catalyst loading is 4.0 mg cm$^{-2}$ for NiFe and 1.0 mg cm$^{-2}$ for Pt/Ru/C. The performance of MTCP-50-based AEMWE was studied fed with 1 M KOH aq. and pure water. The polarization curve at different temperatures (from 30 °C to 90 °C) over an applied voltage range of 1.3–2.0 V was recorded using a LANHE battery test station. EIS was monitored by electrochemical workstation (Zahner Zennium E) at 1.6 V in the frequency from 100 kHz to 1 Hz at different temperatures to evaluate the ohmic resistance. The single-cell durability was conducted at a constant current density of 0.5 A cm$^{-2}$ with 1 M KOH aq. feed at 60 °C and 1.0 A cm$^{-2}$ at 80 °C.

### AEMFC testing

Typically, the MTCP-50 ionomer solution (5 wt%) and Pt/C (Johnson Matthey HiSpec 4000, 40 wt% Pt, for cathode) or Pt/Ru/C (Johnson Matthey HiSpec 10,000, 40 wt% Pt and 20 wt% Ru, for anode) were mixed in an isopropanol/DI water (9 to 1) to prepare catalytic ink. After the ink was evenly distributed via ultrasonic treatment for 1 h, it was sprayed onto MTCP-50 membrane with both loadings of 0.2 mg cm$^{-2}$. AEMFC performance of $I$-V curve was tested using Scribner 850e Fuel Cell Test System with 0.2 L min$^{-1}$ H$_2$/O$_2$ flow rate at 75 °C or 90 °C and 0.1 MPa back pressure. The open-circuit voltage durability test of the MTCP-50-based AEMFC was operated at 60 °C, and gas flow rate 0.1/ 0.1 Lmin$^{-1}$ of H$_2$/O$_2$.

## Data availability

The authors declare that the data supporting the findings of this study are available within the paper, Supplementary Information and Source Data files. Further data beyond the immediate results presented here are available from the corresponding authors upon request. Source data are provided with this paper.

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

## Acknowledgements

This research was supported by the National Key R&D Program of China (No. 2020YFB1505601, 2021YFB4000302), the National Natural Science Foundation of China (Nos. 22038013, 21720102003, 22278388, 12275270), the Fundamental Research Funds for the Central Universities (20720220007).

## Author contributions

W.J.S. designed the experiments, fabricated the membranes, performed the membrane characterizations, prepared the MEA and conduced corresponding tests, collected date. K.P tested the performance of AORFBs. W.X., H.J.Z., and B.J.Y. measured the PALS of the membranes. X.L. helped with AEMWEs testing. H.Q.Z. and X.L. helped with AEMFCs testing. X.L.G., L.W., T.W.X., supervised and guided the work. W.J.S. and X.L.G. discussed all experimental data and wrote the manuscript. L.W., T.W.X., Z.J.Y. helped revised the manuscript. All authors contributed to the data analysis.

## Competing interests

The authors declare no competing interests.
