## [Peer review file · Nature Communications]

REVIEWER COMMENTS

Reviewer #1 (Remarks to the Author):

The present manuscript describes the synthesis of a family of advanced anion-exchange membranes exhibiting an outstanding level of performance and durability, which are demonstrated suitable for implementation in a wide variety of electrochemical energy conversion and storage devices such as redox flow batteries, water electrolyzers and anion-exchange membrane fuel cells.

Despite the outstanding results displayed in the manuscript, the latter simply packs too much information in too little space; the general reader is unable to read through the details and appreciate the depth of the scientific progress obtained by the proposed family of materials. It is necessary to rearrange the subject-matter of the manuscript and split it in no less than two or three manuscripts. The first manuscript must disclose the synthesis of the ionomers, their formation into a membrane, the advanced physicochemical characterization (that must elucidate the details of the interplay between phase separation and the mechanism for the long-range conduction of ions) and finally the implementation of the “best” membrane into one type of electrochemical energy conversion and storage device, displaying outstanding performance and durability. It is crucial that the authors benchmark the results with the best data found in the literature on competing materials. This first manuscript, more fundamental in nature, would be a perfect match for Nature Communications.

Additional results detailing the implementation of the proposed AEMs and ionomers into one or more distinct electrochemical energy conversion and storage devices must be then reported in a distinct manuscript. In such manuscript(s) the authors must disclose in detail the fabrication of the devices and their electrochemical testing for performance and durability. Once again, benchmarking is crucial and, if at all possible, it should be carried out by the authors themselves.

This arrangement of the information available to the authors would be much more accessible to the general reader and would be a much better way to disseminate to the scientific community the advancements achieved, improving the balance between impact and scientific detail.

For all the reasons above this manuscript is not suitable for publication on Nature Communications and must be rejected.

Reviewer #2 (Remarks to the Author):

The paper reports an optimized AEM, its scaled-up fabrication, membrane properties, and the performance in fuel cells, electrolyzers and redox flow batteries.

It is pity that authors do not refer to this work, which seems to be quite relevant:

Mayadevi, T. S., Seounghwa Sung, Listo Varghese, and Tae-Hyun Kim. 2020. "Poly(meta/para-Terphenylene-Methyl Piperidinium)-Based Anion Exchange Membranes: The Effect of Backbone Structure in AEMFC Application" *Membranes* 10, no. 11: 329.

<https://doi.org/10.3390/membranes10110329>

"The m-p-MP-50, having 50 mol% of the meta-terphenyl unit, exhibited the highest conductivity while experiencing the least swelling together with the highest mechanical stability among three m-p-MP-ys with different compositions. This is due to the well-defined morphology resulting from the controlled free volume, which in turn originates from the balanced conformational structure between the linear para-terphenyl and kink-structured meta-terphenyl units."

A small difference to the cited work is the crosslinking by vinylbenzylchloride.

Nevertheless, the work shown in this paper is impressive. The same material was used in three different applications. The performances are excellent. The alkaline stability is exciting. It is very nice to see that membrane production reached demonstration scale.

Therefore, I think the work can be published in this journal after corrections are made.

Comments

1. Please mention/discuss the above mentioned work (Mayadevi et al.)
2. Perhaps the title of the paper can be edited, to stress more the novelty, for example: "Upscaled production of an optimized piperidinium-based AEM and its long-term operation in fuel cells, electrolyzers and flow batteries"
3. Line 68: polybenzimidazoles are quite alkaline stable, I would not list them as easily degradable.
4. PALS shows pore diameters of 4-5 Angstrom, while the diameter of hydrophilic domains is 10 times larger. I believe the effect of the free volume as measured by PALS is not really relevant for the conductivity, unless we assume that the hydrophilic domains are not perfectly connected, i.e. there is a bottle neck at some point, in the dimension of the free volume. Then the dimension of the free volume by PALS is important, because it is a limiting factor. For example, the Gierke model for Nafion shows 4 nm domains, and 1 nm connecting phases. Is it the connecting phases which you see by PALS? Why not the domains, is the life time of positrons so limited that they cannot reach the wall of a domain? Please consider these things when revising the manuscript.
5. You have several Figures showing the stability or durability of membranes versus something, for example Fig 2i, 4h, S32. (a) Please show the references for the data in the SI, or refer to the table in the

SI in the Figure caption. (b) Please clarify what stability means – was the test stopped after this time, even though the membrane was still OK (e.g. preset 100 h or 1000 h test time), or did the membranes fail – then what is the definition of failure, broken into pieces, >10% conductivity loss,?

6. L174/L175: why does rotation lead to a tightly stacked polymer structure and thus to large swelling? I would expect that the high chain mobility results in less chain/chain interactions and thus in more swelling. Increased chain-chain interaction (stacking) should counteract the swelling.

7. The alkaline stability is impressive. Quite similar polymers like Versogen's Piperion showed loss of conductivity in other people's work (Membranes 2022, 12, 989. <https://doi.org/10.3390/membranes12100989>). I am wondering why. Most researchers use KOH, because KOH is more soluble than NaOH and has higher conductivity than NaOH and thus is preferred in electrochemical devices. Is NaOH less aggressive than KOH? Why? Which role plays the cation in alkaline degradation? On another line, glass slowly dissolves in alkaline solutions – did you refresh the solution or use the same solution for 300 days, and would degradation of glass change the hydroxide concentration? Please consider these points in your revised manuscript.

8. L210/L211: CE should increase when current density increases. Probably your values are overly exact and it would be better to display 99.6% instead of 99.61% and 99.56%.

9. Figure 3g: Please state which current density the data is for, EE varies a lot with current density.

10. L284: The opposite trend is observed in Figure 4c and 4f, the resistance is lower in 1M KOH than in DI water.

11. Conductivity measurement: Please report how you prevented contamination with CO₂ from air. For example, it is possible to purge the carbonates electrochemically to get the true hydroxide conductivity.

12. SI L66: Please show how you use the slope in that equation

13. LSI L69: Is 54 um wet or dry thickness? At another point, you use 28 um thick membrane. Please report the membrane thickness.

14. It would be nice to see the gel content, to prove that crosslinking occurs during drying at 80C, and that the observed differences are not related simply to the additional hydrophobic groups.

15. Please add standard deviations for mechanical properties, and report the temperature and rh during measurement (if known)

16. SI Fig 12: WU increases from 40 to 70%, but swelling remains quite same, 7-10, 10-15%. I did not calculate this, but it seems that this cannot be only the effect of pores. The swelling may show anisotropic behavior, which is quite common; thickness swelling could be larger than length swelling. (a) please report the thickness swelling as well (b) you produced the membrane in a roll-to-roll process. Is there a difference between machining and transverse direction? Then, is length x or y direction, and did you strictly observe the orientation when you cut out samples? (c) based on the above, please edit the discussion below Fig 12.

17. Fig S16: Please remove the protons from the ammonium ions – in alkaline conditions, it should be the amine. And please correct the missing double bond in the phenyl ring, also in Fig 1.

18. Fig S18: such as OH^{*}, H^{*}: In alkaline conditions, hydroxyl radicals are expected to rapidly degrade, and we may see mainly superoxide radicals (ChemSusChem, 10 (2017) 3056–3062. <https://doi.org/10.1002/cssc.201700760>)

19. SI L515: “best barrier properties” – for that you need a comparison, but it seems you only report data for one material

20. Table S4: PBI-OO, ref 14: PBI-OO is not the AEM in that work, but a spiro-ionene

21. The scaled-up production is exciting. Please discuss this more in detail – were adjustments necessary, like changes in monomer concentration, temperature control, time,?

Minor errors/typos/language

1. Line 59: high-selectively: highly selective

2. L63: Although..., they still... has to be one sentence

3. Significantly facilitate?

4. L84: networks

5. Fig 1: triphenyl

6. L124: characterize

7. Roman (also in Fig S14)

8. Can be reached with almost unattenuated. – please rephrase

9. Additionally beneficial – the additional benefit of

10. Cooper

11. L441: Pu

12. Fig S27: cycle

Response to reviewers' comments

We are grateful to all reviewers for their time and constructive comments. Below are the point-to-point response.

Reviewers' comments (blue); Authors' response (black); Revised text in manuscript and ESI (yellow highlight).

Reviewer #1 (Remarks to the Author):

The present manuscript describes the synthesis of a family of advanced anion-exchange membranes exhibiting an outstanding level of performance and durability, which are demonstrated suitable for implementation in a wide variety of electrochemical energy conversion and storage devices such as redox flow batteries, water electrolyzers and anion-exchange membrane fuel cells.

Despite the outstanding results displayed in the manuscript, the latter simply packs too much information in too little space; the general reader is unable to read through the details and appreciate the depth of the scientific progress obtained by the proposed family of materials. It is necessary to rearrange the subject-matter of the manuscript and split it in no less than two or three manuscripts. The first manuscript must disclose the synthesis of the ionomers, their formation into a membrane, the advanced physicochemical characterization (that must elucidate the details of the interplay between phase separation and the mechanism for the long-range conduction of ions) and finally the implementation of the "best" membrane into one type of electrochemical energy conversion and storage device, displaying outstanding performance and durability. It is crucial that the authors benchmark the results with the best data found in the literature on competing materials. This first manuscript, more fundamental in nature, would be a perfect match for Nature Communications.

Additional results detailing the implementation of the proposed AEMs and ionomers into one or more distinct electrochemical energy conversion and storage devices must be then reported in a distinct manuscript. In such manuscript(s) the authors must disclose in detail the fabrication of the devices and their electrochemical testing for performance and durability. Once again, benchmarking is crucial and, if at all possible, it should be carried out by the authors themselves.

This arrangement of the information available to the authors would be much more accessible to the general reader and would be a much better way to disseminate to the scientific community the advancements achieved, improving the balance between impact and scientific detail.

For all the reasons above this manuscript is not suitable for publication on Nature Communications and must be rejected.

Response: We thank the reviewer for offering valuable and profound comments. We have added more information in the revised manuscript, including the interplay between phase separation and ion conduction in Fig. R1 (Fig. 2 in the revised manuscript), and the relationship between membrane structure and properties in Fig. R2 (Fig. 3 in the revised manuscript). Additionally, we performed the solid-state nuclear magnetic resonance (ssNMR) to reveal the motion strength of ions by the analysis of

spin-spin relaxation time (T_2). As shown in Fig. R3 (Fig. S9 in the revised supplementary information), the interior ^{35}Cl in MTCP-50 displays narrower signal and a longer T_2 (Fig. R2a) and than the other membranes, which indicates the less confined motion in the MTCP-50 membrane. The resulting fast anion transport behavior in MTCP-50 may be attributed to the more interconnected voids and more-ordered ion domains.

As for the reviewer's suggestion of splitting the manuscript, we first sincerely thank the reviewer for the guidance on the layout of our manuscript. The original intention of our membrane structure design is to construct microporous polymers based on a chemically stable strategy while achieving the improvement of conductivity, stability, and selectivity. We deem that a single application cannot fully disclose the various properties of the membrane. For example, in the aqueous redox flow batteries, the focus is on ionic conductivity and selectivity, while the chemical stability of the membrane is not emphasized. But for water electrolyzers and fuel cells which work in an alkaline environment, more attention is paid to conductivity and chemical stability. Therefore, we consider that not splitting the manuscript can better reflect the advantages of our membrane structure design and be more reasonable. Additionally, it is the first AEM that can simultaneously display outstanding performance and long-term durability in energy storage and conversion devices represented by aqueous redox flow batteries, water electrolyzers, and fuel cells. We believe that the large-scale production and diversified applications of the membrane will further promote membrane marketization and the progress of energy/electrochemical community, which is the great significance of our scientific research.

The major revisions were highlighted in the revised manuscript: Fig. 2, Fig. 3.

Fig. R1 Characterization of MTCP-x AEMs. a-c, Computational modeling results of the pore surfaces for the MTCP-x series applying a 2.0 Å probe diameter; teal and magenta indicate the accessible (interconnected) and non-accessible (disconnected) surface area, respectively. d, Pore size distributions obtained from PALS using CONTIN analysis. e, The CO₂ adsorption isotherms of MTCP-x AEM at 298.15 K. f, SAXS patterns of MTCP-x. g-i, AFM microphase morphology of MTCP-x (the inner images are the distribution of hydrophilic phase width of MTCP-x analyzed by the Nano Measurer).

Fig. R2 Membrane relevant properties. **a**, The analysis of T_2 from the ^{35}Cl ss-NMR of MTCP-x. **b**, Temperature-dependent OH^- conductivity of MTCP-x. **c**, Water uptake and dimensional swelling versus temperature. **d**, Alkali stability with respect to OH^- conductivity of MTCP-50 in 1 M NaOH at 80 °C. **e**, Summary of the relationship between OH^- conductivity and alkaline stability (recently research progress for AEMs of OH^- conductivity over 90 mS cm^{-1} at 80 °C and ex-situ alkaline stability exceeding 500 h. Refer to the table in the Supplementary Table 4). **f**, TS and EB of MTCP-x in OH^- form at room temperature (dry state).

Fig. R3 ^{35}Cl solid state NMR (ss-NMR) measured for MTCP-x.

Reviewer #2 (Remarks to the Author):

The paper reports an optimized AEM, its scaled-up fabrication, membrane properties,

and the performance in fuel cells, electrolyzers and redox flow batteries. It is pity that authors do not refer to this work, which seems to be quite relevant: Mayadevi, T. S., Seounghwa Sung, Listo Varghese, and Tae-Hyun Kim. 2020. "Poly(meta/para-Terphenylene-Methyl Piperidinium)-Based Anion Exchange Membranes: The Effect of Backbone Structure in AEMFC Application" *Membranes* 10, no. 11: 329. <https://doi.org/10.3390/membranes10110329>. "The m-p-MP-50, having 50 mol% of the meta-terphenyl unit, exhibited the highest conductivity while experiencing the least swelling together with the highest mechanical stability among three m-p-MP-ys with different compositions. This is due to the well-defined morphology resulting from the controlled free volume, which in turn originates from the balanced conformational structure between the linear para-terphenyl and kink-structured meta-terphenyl units.". A small difference to the cited work is the crosslinking by vinylbenzylchloride. Nevertheless, the work shown in this paper is impressive. The same material was used in three different applications. The performances are excellent. The alkaline stability is exciting. It is very nice to see that membrane production reached demonstration scale. Therefore, I think the work can be published in this journal after corrections are made.

Response: We thank the reviewer for positive feedback and insightful comments to further improve the quality of our manuscript. We have addressed these comments point by point in the following paragraphs.

Q1. Please mention/discuss the above mentioned work (Mayadevi et al.)

Response: The mentioned job has been cited in the revised manuscript. It should be noted that there exists much difference between the reference and this job. (1) Differences in structure and membrane-relevant properties. For our membrane, the crosslinking strategy was introduced into the structural design. The crosslinking structure dramatically reduced the swelling ratio and enhanced the stability of AEMs. (2) A better understanding of structure-property relationships of fabricated AEMs. Variety characterizations, including PALS, CO₂ adsorption coupled with molecular simulation were performed to uncover the ultramicropores existing in membrane and their variation with monomer proportion. It enables readers and us to have a deeper understanding of the structure's effects on performance. (3) The universality of AEM applications and breakthroughs in performance of various electrochemical devices. Our high-performing AEM can be suitable in a variety of electrochemical devices (flow batteries, water electrolyzers, and fuel cells) and achieve high performance and durability, which is impossible for most reported AEMs.

Q2. Perhaps the title of the paper can be edited, to stress more the novelty, for example: "Upscaled production of an optimized piperidinium-based AEM and its long-term operation in fuel cells, electrolyzers and flow batteries"

Response: We agree with your suggestion. After our in-depth consideration, we think the title edited as "Upscaled production of an ultramicroporous piperidinium-based AEM enables long-term operation in fuel cells, electrolyzers and flow batteries" is perhaps more appropriate.

Q3. Line 68: polybenzimidazoles are quite alkaline stable, I would not list them as easily degradable.

Response: We agree with you that polybenzimidazole polymers are alkaline stable. To avoid misunderstanding, we removed the polybenzimidazoles and polyolefins in the revised manuscript.

Q4. PALS shows pore diameters of 4-5 Angstrom, while the diameter of hydrophilic domains is 10 times larger. I believe the effect of the free volume as measured by PALS is not really relevant for the conductivity, unless we assume that the hydrophilic domains are not perfectly connected, i.e. there is a bottle neck at some point, in the dimension of the free volume. Then the dimension of the free volume by PALS is important, because it is a limiting factor. For example, the Gierke model for Nafion shows 4 nm domains, and 1 nm connecting phases. Is it the connecting phases which you see by PALS? Why not the domains, is the life time of positrons so limited that they cannot reach the wall of a domain? Please consider these things when revising the manuscript.

Response: The hydrophilic domains provide ion-conducting channels, which is a widely accepted opinion. We have supplemented the discussion about hydrophilic domains in the revised manuscript (Lines 150-166). But there's no denying that dense and hydrophobic polymer segments inevitably block ions as they transport in the membrane, which greatly reduces the ion transport rate. Much research shows that the increase of free volume promotes the enhancement of ionic conductivity (*Angew. Chem. Int. Ed.* 2016, 55, 11499–115. <https://doi.org/10.1002/anie.201605916>; *Chemical Engineering Journal* 418 (2021) 129311. <https://doi.org/10.1016/j.cej.2021.129311>). We performed the PALS to probe and analyze the pore size, and further used CO₂ adsorption to verify the connectivity of free volume. These characterization results demonstrated that the conductivity is positively correlated with the well-defined pore. The large, evenly distributed, and well-connected free volumes inside AEM enable higher conductivity by weakening the obstruction of ion transport arising from the dense packing of polymer segments. We do not exclude the reviewer's assumption that the free volume may connect the hydrophilic domains. Generally speaking, more and well-connected free volumes will increase the connect probability of hydrophilic domains. However, regretfully, we cannot determine the connection phase by PALS. PALS is a method to study the nano-scaled free-volume holes in polymers. The o-PS will preferentially locate within the pore space and annihilate immediately upon collision with the free-volume void walls. The lifetime for o-PS usually ranges from 1 ns to 10 ns. However, the hydrophilic domains are formed by the aggregation of ionic groups rather than free volume holes inherent in AEM. Therefore, PALS can probe all holes within the AEM but cannot detect the hydrophilic domains.

Q5. You have several Figures showing the stability or durability of membranes versus something, for example Fig 2i, 4h, S32. (a) Please show the references for the data in the SI, or refer to the table in the SI in the Figure caption. (b) Please clarify what stability means – was the test stopped after this time, even though the membrane was still OK

(e.g. preset 100 h or 1000 h test time), or did the membranes fail – then what is the definition of failure, broken into pieces, >10% conductivity loss,?

Response: We appreciate your comments and helpful suggestions. (a) We have added the sentence “Refer to the table in the supplementary information” for each comparison figure (including Fig 3e, 5d, 5h, 6c, S32 and S33) in the revised Figure caption. (b) We describe below the stop time of alkali stability and performance operating stability tests, respectively. For the alkaline stability measurement, we recorded the alkali aging time for 8000 h, which was ended before preparing the manuscript. But, it should be noted that the MTCP-50 AEM is still transparent and flexible and remains ~80% mechanical toughness and 94.3% hydroxide conductivity, which indicates the AEM didn't fail. The test can be continued. For the cycling test of AORFB, the performance was conducted at high electrolyte concentrations of 1.5 M and 2 M, respectively. It must be pointed out that the cycling time of most reported works is about 200 h, and they almost run at low concentrations (≤ 0.5 M). The cycling time in our work of more than 500 h at such high concentration is already better than recent advanced level. Thus longer time is not continued, limited by the shortage of equipment and other work needed to be carried out. For the durability test of AEMWE, we firstly performed a 2500 h test at the prevailing current density of 0.5 A cm^{-2} . This more prolonged test operation accompanied by extremely low voltage degradation can already demonstrate the advancement of our MTCP-50 AEM in current AEMWE. Then we switched to the test to more harsh operating conditions (high temperature of $80 \text{ }^\circ\text{C}$ and increased current density to 1 A cm^{-2}). Although the MTCP-50-based AEMWE could maintain stable operation, it was unfortunately terminated after 500 h due to equipment failure. For the open-circuit voltage test of AEMFC, 1000 h is the common test record for most current PEMFC research. So, we presetted 1000 h for evaluating the durability of MTCP-50 AEM-based fuel cells.

Q6. L174/L175: why does rotation lead to a tightly stacked polymer structure and thus to large swelling? I would expect that the high chain mobility results in less chain/chain interactions and thus in more swelling. Increased chain-chain interaction (stacking) should counteract the swelling.

Response: We agree that high chain mobility results in less chain/chain interactions and thus in more swelling. The MTCP-70 with 70 mol% of the m-terphenyl unit possesses more higher flexibility conformation. Thus, the increased flexibility of the polymer chain promotes water absorption, which plasticizes the polymer chain, leading to high chain mobility and large swelling. We have rediscussed this issue in the revised manuscript (Lines 192-195).

Q7. The alkaline stability is impressive. Quite similar polymers like Versogen's Piperion showed loss of conductivity in other people's work (Membranes 2022, 12, 989. <https://doi.org/10.3390/membranes12100989>). I am wondering why. Most researchers use KOH, because KOH is more soluble than NaOH and has higher conductivity than NaOH and thus is preferred in electrochemical devices. Is NaOH less aggressive than KOH? Why? Which role plays the cation in alkaline degradation? On another line, glass

slowly dissolves in alkaline solutions – did you refresh the solution or use the same solution for 300 days, and would degradation of glass change the hydroxide concentration? Please consider these points in your revised manuscript.

Response: We would like to thank the reviewer for their profound comments. (a) According to your comment, we believe the impressive membrane alkaline stability should be attributed to the following three aspects. (1) The flexible m-triphenyl on polymer backbone reduce the restriction on piperidinium ring strain relaxation. Previous studies have shown that the distortion of the piperidinium ring conformation imposed by the rigid backbone leads to ionic loss, and the ionic loss increases with the rigidity of the backbones (*Adv. Funct. Mater.* 2018, 28, 1702758. <https://doi.org/10.1002/adfm.201702758>). Therefore, compared with the structure fully composed of rigid p-triphenyl, the introduction of partial flexible m-triphenyl in polymer backbone may reduce the restriction on ring strain relaxation and enables the improvement of membrane stability. (2) The steric hindrance of cross-linked structure within MTCP-50 reduced the chance of hydroxide attack, improving the alkaline stability of AEM (*Energy Environ. Sci.*, 2017, 10, 275. <https://doi.org/10.1039/C6EE03079C>). (3) Increased water molecules hindered the nucleophilicity and basicity of hydroxide. The increased flexibility of the polymer backbone promotes water absorption of the membrane. The WU of MTCP-50 (80% at 80 °C) is higher than Piperion (PAP-TP-85, 63% at 80 °C), which may affect the membrane stability by diluting the concentration of hydroxide ions toward the nucleophilic attack (*J. Mater. Chem. A*, 2019, 7, 15895–15906. <https://doi.org/10.1039/C9TA05531B>). Relevant discussions have been added to the revised manuscript (Lines 208-211).

(b) We agree with you that the KOH has an advantage over NaOH. Up to now, there are still no universal test protocols for *ex-situ* alkaline stability. In order to answer the reviewer's question, we tested the alkali stability of the MTCP-50 AEM in different alkali solutions (1 M NaOH and 1 M KOH at 80 °C) simultaneously. As shown in Fig. R4, after one-month alkali aging at 80 °C, the conductivity loss of MTCP-50 is 1.71% in 1 M NaOH and 1.78% in KOH. According to our comparative experimental results, the influence of alkali solution type on membrane stability is slightly different, but the difference is not obvious. NaOH appears to be less aggressive than KOH, and a recently reported study about the simulation of alkaline electrolyte diffusion in membranes shows some possible reasons. One part, the attractive interactions between the quaternary ammonium ions and OH⁻ ions in K⁺ solution are stronger than those in Na⁺ solution. Another part, Na⁺ is easier to agglomerate than K⁺. Affected by the attraction of metal ions to OH⁻ ions, the agglomerates of OH⁻ are easier formed in NaOH, which reduces the dispersibility of OH⁻ in water. So, there are more OH⁻ ions around the quaternary ammonium ions in the KOH solution (*Science China Technological Sciences*, 63, 2241–2255 (2020). <https://doi.org/10.1007/s11431-020-1615-0>).

Fig. R4 Alkaline stability of MTCP-50 soaking in different alkali solutions (1 M NaOH and 1 M KOH) at 80 °C.

(c) In our long alkaline test, we refreshed the alkaline solution and glass bottle once a month to avoid the influence of concentration change on the membrane stability. We stated this point in our revised manuscript (Line 459).

Q8. L210/L211: CE should increase when current density increases. Probably your values are overly exact and it would be better to display 99.6% instead of 99.61% and 99.56%.

Response: As suggested, we have corrected this point in the revised manuscript. Please see Lines 244 and 245.

Q9. Figure 3g: Please state which current density the data is for, EE varies a lot with current density.

Response: All of the EE comparisons in Fig. 3g (Fig. 4g in the revised manuscript) are at the same current of 100 mA cm⁻². We have stated this point in the revised manuscript (Line 280).

Q10. L284: The opposite trend is observed in Figure 4c and 4f, the resistance is lower in 1M KOH than in DI water.

Response: We are sorry that the absence of discussion of Fig. 4c may have caused your misunderstanding. We performed the *I-V* curves and EIS spectra in 1 M KOH (Fig. 5c in the revised manuscript) and pure water fed (Fig. 5f in the revised manuscript), respectively. Undoubtedly, the resistance is lower in KOH system than in pure water. Herein, to avoid ambiguity and deliver a more accurate description, the ohmic resistance values of different liquid systems are discussed separately in the revised manuscript. Please see Lines 304 and 318.

Q11. Conductivity measurement: Please report how you prevented contamination with CO₂ from air. For example, it is possible to purge the carbonates electrochemically to get the true hydroxide conductivity.

Response: During our testing process, we minimized the exposure time of the membrane in the air as much as possible to reduce the effects of carbon dioxide on the

hydroxide conductivity. Besides, we did not do other carbon dioxide removal measures. We will pay more attention to reducing the effect of carbonate species on hydroxide conductivity in the subsequent work.

Q12. SI L66: Please show how you use the slope in that equation

Response: The slope value in the equation is P , calculated from the permeability measurement. It represents the permeability coefficient and reflects the barrier property of the membrane to redox-active molecules.

Q13. LSI L69: Is 54 μm wet or dry thickness? At another point, you use 28 μm thick membrane. Please report the membrane thickness.

Response: The membrane with a thickness of 54 μm is dry. And the MTCP-50 AEM with a thickness of 54 μm was used for AORFB. In the application of AEMWE and AWEFC, the AEM thickness was ~ 25 μm .

Q14. It would be nice to see the gel content, to prove that crosslinking occurs during drying at 80°C, and that the observed differences are not related simply to the additional hydrophobic groups.

Response: We agree that gel content would be helpful to prove the occurrence of membrane crosslinking. We have performed the situ FT-IR to verify the in-situ crosslinking process of terminal vinyl groups under thermal initiation. Done as suggested, we performed the solubility and gel content test of MTCP-x AEMs and added them to the revised manuscript. Firstly, we take MTCP-50 as an example to prove the occurrence of crosslinking through a solubility experiment. One membrane was obtained by casting MTCP-50 solution at 30 °C (MTCP-50@30 °C) and the other at 80 °C (MTCP-50@80 °C). The two membranes were soaked in NMP at 60 °C. We recorded the dissolution process with photographs. As shown in Fig. R5, the MTCP-50@30 °C is completely dissolved after 1.5 h, and the solution is transparent and faint yellow. The MTCP-50@80 °C remains intact, and the solution is colorless and transparent. After 6 h, The yellow color of solution MTCP-50@30 °C deepens. While the MTCP-50@80 °C is mostly dissolved, with some insoluble fragments. Even after 24 h, MTCP-50@80 °C still has some insoluble matter and is deepened with solution yellow. This proves that crosslinking occurs during drying at 80 °C. And our cross-linked membrane is moderately soluble because of the low grafted degree of cross-linked groups. The gel content of the cross-linked MTCP-x AEMs were measured by immersing the thoroughly dried MTCP-x samples in NMP solution at 60 °C for 1 h followed by drying and weighing. The gel content was calculated from the ratio of the dried sample weight after the test to its initial weight.

Fig. R5 The solubility experiment photographs of MTCP-50 membranes (1 and 2 represent MTCP-50@30 °C MTCP-50@80 °C, respectively).

Table R1 Gel content of MTCP-x AEMs

AEMs	MTCP-30	MTCP-50	MTCP-70
Gel content (%)	93.54 ± 0.4	93.50 ± 0.5	93.57 ± 0.5

Q15. Please add standard deviations for mechanical properties, and report the temperature and rh during measurement (if known).

Response: As the review suggested, we repeated the test of the mechanical properties and added the standard deviations to mechanical properties in the revised manuscript. Please see Fig. R6 and 7 (Fig. S10 and S18 in the revised supplementary information). All stress-strain curve measurements were performed at room temperature without any additional conditions.

Fig. R6 Mechanical properties of MTCP-x in OH⁻ form **at room temperature** (dry state).

Fig. R7 The variation of mechanical properties (in OH⁻ form, room temperature and dry state) of MTCP-50 in 1 M NaOH at 80 °C over 8000 h.

Q16. SI Fig 12: WU increases from 40 to 70%, but swelling remains quite same, 7-10, 10-15%. I did not calculate this, but it seems that this cannot be only the effect of pores. The swelling may show anisotropic behavior, which is quite common; thickness swelling could be larger than length swelling. (a) please report the thickness swelling as well (b) you produced the membrane in a roll-to-roll process. Is there a difference between machining and transverse direction? Then, is length x or y direction, and did you strictly observe the orientation when you cut out samples? (c) based on the above, please edit the discussion below Fig 12.

Response: We agree with the reviewer that membrane show anisotropic behavior. (a) According to the reviewer's comments, we have measured the thickness swelling ratio, as shown in Fig. R8. The data has been added to Supplementary Figure 13c.

Fig. R8 Temperature-dependent thickness swelling properties of MTCP-x AEMs.

(b) We adopted the solution casting method for producing the continuous membranes. The membrane samples with 4 × 4 cm strips for measurement show the same length swelling ratio in x or y directions. Thus, there is no difference between machining and transverse direction.

(c) According to the reviewer's suggestions, we rediscussed below Fig. S12 (Fig. S13 in the revised supplementary information).

Q17. Fig S16: Please remove the protons from the ammonium ions – in alkaline conditions, it should be the amine. And please correct the missing double bond in the phenyl ring, also in Fig 1.

Response: Done as suggested. Please see Fig.1 and Supplementary Figure 17 in the revised text.

Q18. Fig S18: such as OH*, H*: In alkaline conditions, hydroxyl radicals are expected to rapidly degrade, and we may see mainly superoxide radicals (ChemSusChem, 10 (2017) 3056–3062. <https://doi.org/10.1002/cssc.201700760>)

Response: Thank you for your concerns. The excellent alkali stability of the AEM can be evidenced by the data in alkaline solution. The intention of the test using the Fenton reagent is to verify the oxidation stability of the AEM. This is a widely adopted method in current research (*Adv. Mater.* e2210432 (2023). <https://doi.org/10.1002/adma.202210432>).

19. SI L515: “best barrier properties” – for that you need a comparison, but it seems you only report data for one material

Response: We thank you for the comment. Unfortunately, H₂ permeability values cannot be directly compared due to different test methods and conditions. For more accurate expression, we changed “best” to “excellent” in the revised Supporting information.

Q20. Table S4: PBI-OO, ref 14: PBI-OO is not the AEM in that work, but a spiro-ionene.

Response: Thank you for pointing this out. We have corrected this error in the revised Supporting information. Please see Supplementary Table 4. Because of the lack of conductivity data of spiro-ionene, we removed this membrane and added a newly reported FPAP-3 in Fig. 3e of the revised manuscript.

21. The scaled-up production is exciting. Please discuss this more in detail – were adjustments necessary, like changes in monomer concentration, temperature control, time,?

Response: In the pilot-scale synthesis, the reactor was enlarged, the stirring was uneven, and the heat at different positions was unbalanced. So, adjustments from the laboratory level to the pilot scale were necessary. For pilot scale synthesis, we made some optimization of the synthesis parameters. We discuss in detail through the following aspects. (1) We didn't change the monomer concentration in the polymerization step but increased the monomer concentration by 1.5 times (almost approaching the polymer's dissolution limit) in the functionalization step. (2) For temperature control, unlike laboratory-scale reactions that cool with ice baths, the heat transfer effect of pilot-scale reactions is poor, especially with the increase of viscosity during the polymerization process. In our polymerization amplification experiment, an additional cryo cycle device was required to maintain the reaction solution at 0 °C. (3) For the synthesis time, the pilot scale was longer than the laboratory scale during the polymerization step. In laboratory-scale synthesis, we used magnetic stirring, and the viscosity continued to increase during polymerization until the magneton could no

longer rotate. It's considered the reaction endpoint. The time was about 9 h. While for pilot-scale synthesis, mechanical stirring was adopted to make the mixing more uniform. Similarly, the reaction endpoint was when the stirring paddle cannot rotate, but this time was about 24 h.

Minor errors/typos/language

1. Line 59: high-selectively: highly selective;
2. L63: Although..., they still... has to be one sentence;
3. Significantly facilitate?
4. L84: networks;
5. Fig 1: tripbiphenyl;
6. L124: characterize;
7. Roman (also in Fig S14);
8. Can be reached with almost unattenuated. – please rephrase;
9. Additionally beneficial – the additional benefit of;
10. Cooper;
11. L441: Pu;
12. Fig S27: cycle.

Response: All the minor errors above were checked and revised with the suggestions.

REVIEWERS' COMMENTS

Reviewer #2 (Remarks to the Author):

The authors commented well and conducted several additional experiments. Authors did not follow the suggestion of reviewer 1 to split the paper into 3, but I agree with the authors that there is a value in showing performances in fuel cell, electrolyser and RFDB with exactly the same membrane produced in an upscaled process. After a few very minor corrections, the manuscript can be published.

1. I think Maydevi et al deserve more acknowledgement. Please mention that work clearly in the introduction and results and discussion part. Same as Maydevi et al, you found that 50% is the optimized value. You advance the field by crosslinking, and crosslinking does not shift the optimized value of 50%.
2. Reviewer 2 comment 5b: There seems to be a misunderstanding. Figures 2, 3, S32 show graphs of alkaline stability (hours) versus something. Some data points are exactly located at 1000 hours, some exactly at 2000 hours. This somehow indicates that others could have higher stability values, if they would have tested for 3000 or 5000 hours, or maybe even 10,000 hours, exceeding your stability. They just did not test for such a long time. Please mention in the Figure caption that stability is not necessarily end of life data but could be also end of test data.
3. Reviewer 2 comment 10: I am irritated by the expression that MTCP-50 in pure water delivers an excellent performance “attributed to the faster ion transport rate of the membrane itself”. I understood that faster relates to MTCP-50 50 in KOH in comparison to MTCP-50 in DI water. What do you compare when you use “faster”?
4. Comment 13: Please state the thickness values in the manuscript, I could not find them.
5. Comment 15: humidity has an effect on mechanical properties, and may vary on a cold day with room heating (<20% relative humidity) and a warm humid day (> 90%rh). Assuming that all samples were measured on the same day, the trends are correct. But noting the humidity would be good additional information in future work.

Response to reviewers' comments

We unfeignedly appreciate for rigorous but useful advice from the reviews, which is helpful to promote the quality of this work. We have addressed these comments point by point in the following paragraphs.

Reviewers' comments (blue); Authors' response (black); Revised text in manuscript and ESI (yellow highlight).

Reviewer #2 (Remarks to the Author):

The authors commented well and conducted several additional experiments. Authors did not follow the suggestion of reviewer 1 to split the paper into 3, but I agree with the authors that there is a value in showing performances in fuel cell, electrolyser and RFDB with exactly the same membrane produced in an upscaled process. After a few very minor corrections, the manuscript can be published.

Response: Thank you for your positive comments.

Q1. I think Maydevi et al deserves more acknowledgment. Please mention that work clearly in the introduction and results and discussion part. Same as Maydevi et al, you found that 50% is the optimized value. You advance the field by crosslinking, and crosslinking does not shift the optimized value of 50%.

Response: The work of Mayadevi et al. has been mentioned in the revised manuscript. Please see Lines 94-96, 99-100, 181-183, and 186-188.

Q2. Reviewer 2 comment 5b: There seems to be a misunderstanding. Figures 2, 3, S32 show graphs of alkaline stability (hours) versus something. Some data points are exactly located at 1000 hours, some exactly at 2000 hours. This somehow indicates that others could have higher stability values, if they would have tested for 3000 or 5000 hours, or maybe even 10,000 hours, exceeding your stability. They just did not test for such a long time. Please mention in the Figure caption that stability is not necessarily end of life data but could be also end of test data.

Response: We sincerely appreciate for this professional advice. We have done as suggested. Please see Line 246.

Q3. Reviewer 2 comment 10: I am irritated by the expression that MTCP-50 in pure water delivers an excellent performance “attributed to the faster ion transport rate of the membrane itself”. I understood that faster relates to MTCP-50 50 in KOH in comparison to MTCP-50 in DI water. What do you compare when you use “faster”?

Response: We understood the reviewer's meaning and changed “faster” to “fast” in the revised manuscript.

Q4. Comment 13: Please state the thickness values in the manuscript, I could not find them.

Response: We have stated the membrane thickness values in the revised manuscript. Please see Lines 304, 370 and 399.

Q5. Comment 15: humidity has an effect on mechanical properties, and may vary on a cold day with room heating (<20% relative humidity) and a warm humid day (> 90%rh). Assuming that all samples were measured on the same day, the trends are correct. But noting the humidity would be good additional information in future work.

Response: We agree with you that humidity affects mechanical properties. And we will pay attention to the humidity factor in further study.